# A Decade of Lymphoma-Associated Hemophagocytic Lymphohistiocytosis: Does the Outcome Improve?

**DOI:** 10.3390/jcm10215114

**Published:** 2021-10-30

**Authors:** Cheng-Hsien Lin, Yu-Hsuan Shih, Tsung-Chih Chen, Cheng-Wei Chou, Chiann-Yi Hsu, Chieh-Lin Jerry Teng

**Affiliations:** 1Division of Hematology/Medical Oncology, Department of Medicine, Taichung Veterans General Hospital, Taichung 40705, Taiwan; briefsummer@gmail.com (C.-H.L.); rollingstone07@gmail.com (Y.-H.S.); tjchen9477@gmail.com (T.-C.C.); 1983ccwei@gmail.com (C.-W.C.); 2College of Medicine, National Chung Hsing University, Taichung 402202, Taiwan; 3Graduate Institute of Clinical Medicine, College of Medicine, National Taiwan University, Taipei 100233, Taiwan; 4Biostatistics Task Force, Taichung Veterans General Hospital, Taicung 40705, Taiwan; chiann@vghtc.gov.tw; 5Department of Life Science, Tunghai University, Taichung 407224, Taiwan; 6School of Medicine, Chung Shan Medical University, Taichung 40201, Taiwan

**Keywords:** lymphoma, hemophagocytic lymphohistiocytosis, overall survival, performance status

## Abstract

To investigate the potential treatment evolution and outcome improvement, we retrospectively compared clinical characteristics, therapeutic strategies, treatment responses, and overall survival (OS) in patients diagnosed and treated with lymphoma-associated HLH between 2004–2012 (*n* = 30) and 2013–2021 (*n* = 26). Our study showed that the clinical characteristics of lymphoma-associated HLH did not substantially change over the past two decades. However, more patients diagnosed in 2013–2021 were tested for Epstein–Barr virus than those diagnosed in 2004–2012 (69.3% vs. 33.3%; *p* = 0.021). In addition, Eastern Cooperative Oncology Group performance status 3–4 (hazard ratio (HR): 5.38; 95% confidence intervals (CI): 2.49–11.61; *p* < 0.001) and jaundice (HR: 2.91; 95% CI: 1.37–6.18; *p* = 0.006) were poor prognostic factors for lymphoma-associated HLH. With a comparable response rate of lymphoma treatment, patients treated in 2013–2021 had a numerically greater median OS than those treated in 2004–2012 (23.6 ± 19.8 vs. 9.7 ± 4.5 months). However, the difference was not statistically significant (*p* = 0.334). In conclusion, early diagnosis and tailored treatments that balance efficacy and adverse events remain the key to obtaining a better outcome in lymphoma-associated HLH.

## 1. Introduction

Hemophagocytic lymphohistiocytosis (HLH) is a life-threatening hyperinflammatory disorder characterized by uncontrolled immune activation and subsequent organ damage. According to the HLH-2004 diagnostic criteria, patients with HLH can clinically present with fever, splenomegaly, cytopenia, hyperferritinemia, hypofibrinogenemia, hypertriglyceridemia, high soluble CD25, or low NK cell activity. Histopathology studies may reveal hemophagocytosis in the bone marrow, lymph nodes, spleen, liver, and other tissues [1].

Hemophagocytic lymphohistiocytosis can be primary or secondary depending on the underlying etiology. Several genetic mutations and hereditary syndromes related to immune dysregulation have been identified in familial or primary HLH [2]. However, HLH in the majority of adult patients is secondary. Infections, autoimmune diseases, and malignancies are common causes of secondary HLH [3]. Machaczka et al. reported that the estimated annual incidence in Sweden of HLH in adulthood is 0.36/100,000 individuals per year. In addition, HLH occurs in 1.25% of lymphoid malignancies [4]. According to a large retrospective worldwide study, lymphoma is the leading cause of malignancy-associated HLH [3]. T-cell/natural-killer lymphoma, B-cell lymphoma, and Hodgkin lymphoma accounted for 35.2%, 31.8%, and 5.8% of cases, respectively.

Uncontrolled hyperinflammation caused by hyperactivation of CD8-positive T lymphocytes and macrophages, infiltration of inflammatory cells in organs, and excess production of type 1 T-helper cell cytokines is the proposed mechanism of HLH. High serum levels of interferon-gamma, tumor necrosis factor-alpha, interleukin-6, and interleukin-10 are typical laboratory findings [5]. In terms of lymphoma-associated HLH, excess inflammation in the setting of persistent antigen stimulation by cancer cells may play a role, although the precise mechanism remains unclear [1].

The outcome of patients with lymphoma-associated HLH is dismal. We previously reported 30 patients with lymphoma-associated HLH diagnosed between July 2004 and October 2012 [6]. The median overall survival (OS) of these 30 patients was only 231 days. Whether the outcome of lymphoma-associated HLH has improved in recent years with increased awareness and improved therapeutic approaches is unclear. To investigate the potential treatment evolution and outcome improvement in patients with lymphoma-associated HLH over the past decade, we retrospectively compared the clinical characteristics, therapeutic strategies, treatment response, and outcomes of patients diagnosed and treated between 2004–2012 and 2013–2021 in our institution. We also identified the prognostic factors of lymphoma-associated HLH in the current study.

## 2. Methods

### 2.1. Patients 

We retrospectively reviewed the medical records of 59 consecutive patients with lymphoma-associated HLH diagnosed between December 2004 and January 2021 at Taichung Veterans General Hospital, Taiwan. Subtypes of lymphoma were classified according to the World Health Organization classification [7]. In addition, we used the Histiocyte Society HLH-2004 diagnostic criteria to confirm HLH [8]. Since this study focused on patients with non-Hodgkin’s lymphoma (NHL), we excluded three patients with Hodgkin lymphoma-associated HLH. Finally, 56 patients were included in the analyses. The median age of the entire study cohort was 57.5 years (range: 21–83). We further stratified our study population into two groups according to the date of confirmation of lymphoma (2004–2012 (*n* = 30) and 2013–2021 (*n* = 26)). The cutoff date for data analysis was 5 April 2021. Clinical characteristics and outcomes of patients with lymphoma-associated HLH diagnosed in the 2004–2012 group have been reported previously [6]. In the current study, we compared the clinical features and outcomes of two groups of patients with lymphoma-associated HLH, those diagnosed between 2004–2012 and those diagnosed between 2013–2021. This study was conducted in accordance with the principles of the Declaration of Helsinki. The institutional review board of Taichung Veterans General Hospital approved this study and waived the requirement for informed consent because of its retrospective design (No. CE21248B). 

### 2.2. Definitions and Outcome Measurements

We used the Ann Arbor system to stage the NHL [9]. The performance status (PS) was determined based on the Eastern Cooperation Oncology Group (ECOG) scale [10]. We defined jaundice as when the serum total bilirubin of patients was >1.5 mg/dL. The HLH onset was before or after lymphoma was confirmed, one month before or after NHL diagnosis. In addition, HLH occurred simultaneously with lymphoma when the interval between the diagnoses of HLH and lymphoma was within one month. The best response to lymphoma treatment was assessed using the criteria proposed by Cheson et al. [11]. The OS was defined as the period between lymphoma diagnosis and death or when the study was censored. 

### 2.3. Statistical Analyses 

Continuous variables were analyzed using the Mann–Whitney U test as appropriate. Categorical variables were analyzed using the chi-square test or Fisher’s exact test, as required. We used a Cox proportional model to identify lymphoma-associated HLH risk factors, quantified as hazard ratios (HRs) with 95% confidence intervals (CIs). Multivariate analyses further investigated the factors significantly associated with survival using univariate analyses. Furthermore, we used a log-rank test to compare the OS among the different patient groups, as shown by the Kaplan–Meier curves. Statistical significance was set at *p* < 0.05. We used SPSS (version 22.0; SPSS Inc., Chicago, IL, USA) to conduct all statistical analyses.

## 3. Results

### 3.1. Clinical Characteristic Comparisons among Patients Diagnosed in 2004–2012 and 2013–2021

These two groups were comparable in terms of age (*p* = 0.068), sex (*p* = 0.818), ECOG PS (*p* = 0.890), lymphoma stage (*p* = 1.000), international prognostic index (*p* = 0.954), and lymphoma subtype (*p* = 0.427). With regards to laboratory variables, triglyceride (*p* = 0.095), fibrinogen (*p* = 0.424), ferritin (*p* = 0.304), lactate dehydrogenase (*p* = 0.058), disseminated intravascular coagulopathy (*p* = 0.935), and jaundice (*p* = 0.235) were not significantly different between the two groups. Most HLH cases occurred with lymphoma; 70.0% in the cohort diagnosed between 2004 and 2012 and 73.1% in the cohort diagnosed between 2013 and 2021 (*p* = 0.544). Notably, more patients diagnosed in 2013–2021 were tested for Epstein–Barr virus (EBV) than those diagnosed in 2004–2012 (69.3% vs. 33.3%; *p* = 0.021), suggesting that we paid more attention to the impact of EBV on lymphoma-associated HLH over time (Table 1).

### 3.2. Treatment and Outcome Comparison 

With regards to treatments of B-cell lymphoma, a similar percentage of patients diagnosed in 2004–2012 and 2013–2021 received intent-to-cure therapies (92.3% vs. 81.2%; *p* = 0.606). In addition, rituximab was administered to 61.5% and 81.2% of the two cohorts (*p* = 0.406). In terms of T-cell lymphoma, a comparable percentage of patients in these two groups underwent chemotherapy with curative intent (76.5% vs. 60.0%; *p* = 0.415) (Table 2). Eventually, six patients of the study population underwent allogeneic hematopoietic stem cell transplantation (Appendix A). 

We further studied the treatment response and outcomes of patients receiving intent-to-cure therapies. The data showed the complete remission (CR) rate of lymphoma-associated HLH patients diagnosed between 2004–2012 and 2013–2021 was 50.0% and 62.5%, respectively. The treatment responses between these two cohorts were not significantly different (*p* = 0.137) (Table 3).

### 3.3. Overall Survival Comparison 

The overall mortality rate in the entire study cohort was 73.2% (41/56). Only 20.0% and 34.6% of patients diagnosed in 2004–2012 and 2013–2021 remained alive (*p* = 0.353) (Table 3). Since seven of the 56 patients died of fulminant diseases before lymphoma confirmation, this study only investigated the OS time in 49 patients. The median OS of the whole study population was 10.9 ± 3.0 months (Figure 1A). It was 9.7 ± 4.5 and 23.6 ± 19.8 months in the 2004–2012 group (*n* = 27) and 2013–2021 group (*n* = 22), respectively (Figure 1B). Although the median OS time in the 2013–2021 group was numerically greater than that in the 2004–2012 group, the difference was not statistically significant (*p* = 0.334).

### 3.4. Prognostic Factors for Lymphoma-Associated HLH

A univariate analysis showed that ECOG PS 3–4 (HR: 4.19; 95% CI: 2.08–8.42; *p* < 0.001), disseminated intravascular coagulation (HR: 2.06; 95% CI: 1.01–4.19; *p* = 0.046), and jaundice (HR: 2.78; 95% CI: 1.34–5.76; *p* = 0.006) were associated with inferior survival in patients with lymphoma-associated HLH. The multivariate analysis further confirmed these data, showing that ECOG PS 3–4 (HR: 5.38; 95% CI: 2.49–11.61; *p* < 0.001) and jaundice (HR: 2.91; 95% CI: 1.37–6.18; *p* = 0.006) were poor prognostic factors for lymphoma-associated HLH. Remarkably, lymphoma subtypes (*p* = 0.650), ferritin > 3000 ng/mL (*p* = 0.339), and diagnosis period (*p* = 0.336) were not significantly associated with survival (Table 4).

## 4. Discussion 

The current study showed that the clinical characteristics of lymphoma-associated HLH did not substantially change over the past two decades. The ECOG PS 3–4, disseminated intravascular coagulation, and jaundice were poor prognostic factors. With a comparable response rate to lymphoma treatment, patients treated in 2013–2021 had a numerically superior median OS than those treated in 2004–2012, although the difference was not statistically significant. 

The outcome of lymphoma-associated HLH is suboptimal. A multicenter retrospective study by Li et al. [12] demonstrated that the median OS time of patients with lymphoma-associated HLH was only 1.5 months. The catastrophic nature of the disease is one of the potential reasons for this dismal outcome. In our study, mortality occurred in 12.5% (7/56) of the patients before lymphoma treatment initiation. Moreover, more than 90% of the study population was diagnosed with stage III/IV NHL. Importantly, this high incidence of advanced disease did not differ among patients diagnosed in 2004–2012 and 2013–2021 (93.3% vs. 92.3%; *p* = 1.000). Although PET/CT might help identify the possible trigger of HLH and facilitate the early diagnosis of lymphoma [13], early diagnosis remains a challenge for lymphoma-associated HLH. Nonetheless, the OS time of lymphoma-associated HLH numerically increased in the 2013–2021 group relative to the 2004–2012 group (23.6 ± 19.8 vs. 9.7 ± 4.5 months, respectively), even though patients’ characteristics were similar. Increasing awareness and better therapeutic strategies (Appendix A) may be one explanation for this observation because more patients diagnosed in 2013–2021 were tested for EBV than those diagnosed in 2004–2012. 

To combat hyperinflammation caused by lymphoma-associated HLH, our practice routinely used corticosteroids before frontline chemotherapy [14] (La Rosee et al., 2019). Increasing awareness in the past decade might accelerate corticosteroid administration, which potentially diminishes end-organ damage before frontline chemotherapy to the underlying lymphoma. More B-lymphoma-associated HLH identified in 2013–2021 than in 2004–2012 could be another reason for the improved outcome in the past decade. The B-lymphoma-associated HLH accounted for 43.3% and 61.5% of the study population in the 2004–2012 and 2013–2021 groups, respectively (*p* = 0.275). In addition, 61.5% of patients with B-lymphoma-associated HLH in the 2004–2012 group, and 81.2% of these patients in the 2013–2021 group received rituximab treatment. A previous study demonstrated that rituximab diminishes inflammation and reduces the viral load in EBV-associated HLH [15]. Chang et al. [16] further showed that patients with B-cell-lymphoma-associated HLH treated with rituximab had an improved prognosis compared with patients not treated with rituximab. These data confirm the important role of rituximab in HLH treatment, especially in patients with B-cell-associated HLH. 

The optimal chemotherapeutic regimens for lymphoma-associated HLH remain unclear. Compared with the HLH-1994 protocol, the DEP (dexamethasone, etoposide, and cisplatin) regimen significantly improved patients’ overall response rates at weeks 2 and 4. In addition, a higher complete remission rate and a lower recurrence rate at week 4 were observed in patients treated with the DEP regimen [17]. Of note, HLH-1994 was a pediatric study and excluded patients with underlying lymphoma. This comparison might not be entirely fair because controlling the underlying disease is still the cornerstone of lymphoma-associated HLH treatment. Li et al. [12] reported that lymphoma-associated HLH patients who received etoposide and anthracycline had a better OS than patients who did not, suggesting that more intensified regimens containing chemotherapeutic agents effective against lymphoma are crucial to lymphoma-associated HLH. However, not every patient in our study cohort received more intensive frontline treatments than CHOP (cyclophosphamide, adriamycin, vincristine, and prednisolone) or CHOP-like regimens. In fact, our study’s frontline regimens for lymphoma-associated HLH varied and were not significantly different between the two decades compared (Appendix A). Poor PS may be one of the reasons for the limited administration of more intensified frontline therapies to lymphoma-associated HLH. Approximately 40% of our study cohort harbored ECOG PS 3–4 at diagnosis. More importantly, the 2004–2012 and 2013–2021 groups had similar percentages of ECOG PS 3–4 (40.0% vs. 34.6%; *p* = 0.890), suggesting that treating lymphoma-associated HLH in a real-life setting could be much more complicated than that in clinical trials and needs to be tailored to each patient. 

Our study further demonstrated that ECOG PS 3–4, disseminated intravascular coagulation, and jaundice were poor prognostic factors for lymphoma-associated HLH. Various studies have shown different prognostic factors for adult HLH. Hypofibrinogenemia, thrombocytopenia, hypoalbuminemia, hyperferritinemia, and elevated lactate dehydrogenase levels are significant predictors of inferior survival [18,19]. Regardless of these identified prognostic factors, tumor-associated HLH had a markedly worse outcome, emphasizing the importance of underlying disease control in lymphoma-associated HLH. 

The retrospective study design and the limited number of patients were significant limitations of the current study. In addition, we did not routinely check the EBV status in patients with lymphoma-associated HLH, so that we did not analyze the difference of plasma EBV DNA between patients of T/NK cell and B-cell-lymphoma-associated HLH. However, the understanding of the impact of EBV on lymphoma-associated HLH has been growing over time. Prospective and randomized controlled studies with larger patient numbers are needed to confirm our results. In summary, the clinical characteristics of lymphoma-associated HLH did not substantially differ over the past two decades. ECOG PS 3–4, disseminated intravascular coagulation, and jaundice were poor prognostic factors. Increasing awareness and more effective treatments have improved patient survival in the past decade, although the difference did not reach statistical significance. Early diagnosis and tailored treatments that balance efficacy and adverse events remain the key to obtaining a better outcome in lymphoma-associated HLH. 

## Figures and Tables

**Figure 1 jcm-10-05114-f001:**
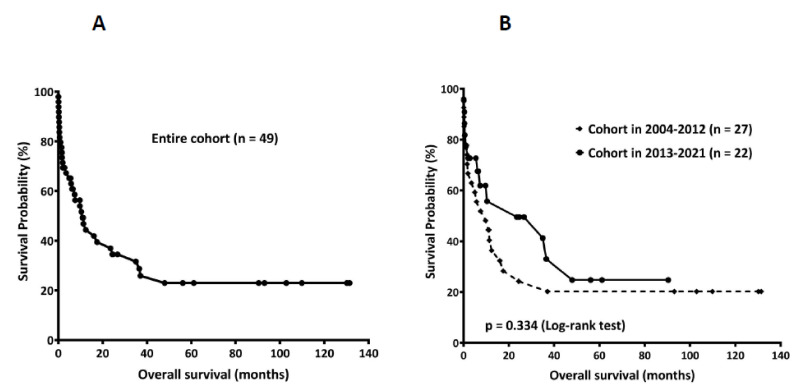
**The overall survival (OS) in patients of lymphoma-associated hemophagocytic lymphohistiocytosis.** (**A**) The median OS of the whole study population (*n* = 49) was 10.9 ± 3.0 months. (**B**) The median OS was 9.7 ± 4.5 and 23.6 ± 19.8 months in the 2004–2012 group (*n* = 27) and 2013–2021 group (*n* = 22), respectively (*p* = 0.334).

**Table 1 jcm-10-05114-t001:** Demographic comparisons among lymphoma-associated HLH patients diagnosed in 2004–2012 and 2013–2021.

	Total (*n* = 56)		Year of Diagnosis 2004–2012 (*n* = 30)		Year of Diagnosis 2013–2021 (*n* = 26)		*p* Value
**Age, median (range, years)**	57.5	(21–83)	51	(21–77)	62	(27–83)	0.068 ^‡^
**Gender, *n* (%)**							0.818 ^†^
Male	26	(46.4%)	13	(43.3%)	13	(50.0%)	
Female	30	(53.6%)	17	(56.7%)	13	(50.0%)	
**Stage, *n* (%)**							1.000 ^#^
Stage 1 & 2	4	(7.1%)	2	(6.7%)	2	(7.7%)	
Stage 3 & 4	52	(92.9%)	28	(93.3%)	24	(92.3%)	
**Cytopenia (≥2 lines), *n* (%)**							0.481 ^#^
No	9	(16.1%)	6	(20.0%)	3	(11.5%)	
Yes	47	(83.9%)	24	(80.0%)	23	(88.5%)	
**Triglyceride, median (range, mg/dL)**	215.5	(69–998)	189	(69–998)	280	(129–511)	0.095 ^‡^
**Fibrinogen, median (range, mg/dL)**	219.8	(60–738)	232	(60–738)	189.5	(81.7–575)	0.424 ^‡^
**Ferritin, median (range, ng/mL)**	5152.9	(500–59,384)	4994.5	(500–34,755)	5250.9	(1185–59,384)	0.304 ^‡^
**LDH, median (range, IU/L)**	1205	(196–7271)	941	(196–7271)	1587	(227–6777)	0.058 ^‡^
**DIC, *n* (%)**							0.935 ^†^
No	38	(67.9%)	21	(70.0%)	17	(65.4%)	
Yes	18	(32.1%)	9	(30.0%)	9	(34.6%)	
**Jaundice, *n* (%)**							0.235 ^†^
No	23	(41.1%)	15	(50.0%)	8	(30.8%)	
Yes	33	(58.9%)	15	(50.0%)	18	(69.2%)	
**ECOG performance status, *n* (%)**							0.890 ^†^
1–2	35	(62.5%)	18	(60.0%)	17	(65.4%)	
3–4	21	(37.5%)	12	(40.0%)	9	(34.6%)	
**IPI, *n* (%)**							0.954 ^†^
0–3	25	(44.6%)	14	(46.7%)	11	(42.3%)	
4–5	31	(55.4%)	16	(53.3%)	15	(57.7%)	
**HLH onset, *n* (%)**							0.544 ^†^
With lymphoma	40	(71.4%)	21	(70.0%)	19	(73.1%)	
Before lymphoma	7	(12.5%)	5	(16.7%)	2	(7.7%)	
After lymphoma	9	(16.1%)	4	(13.3%)	5	(19.2%)	
**EBER, *n* (%)**							0.021 ^†^
Positive	14	(25.0%)	6	(20.0%)	8	(30.8%)	
Negative	14	(25.0%)	4	(13.3%)	10	(38.5%)	
Unknown	28	(50.0%)	20	(66.7%)	8	(30.8%)	
**Lymphoma subtype, *n* (%)**							0.427 ^†^
Diffuse large B-cell lymphoma	29	(51.8%)	13	(43.3%)	16	(61.5%)	
Peripheral T-cell lymphoma	20	(35.7%)	13	(43.3%)	7	(26.9%)	
Angioimmunoblastic T-cell lymphoma	4	(7.1%)	3	(10.0%)	1	(3.8%)	
Anaplastic large cell lymphoma	1	(1.8%)	0	(0.0%)	1	(3.8%)	
NK/T-cell lymphoma	2	(3.6%)	1	(3.3%)	1	(3.8%)	

HLH: hemophagocytic lymphohistiocytosis; LDH: lactate dehydrogenase; DIC: disseminated intravascular coagulation; ECOG: Eastern Cooperative Oncology Group; IPI: international prognostic index; EBER: Epstein–Barr virus-encoded RNAs. Jaundice is defined when serum total bilirubin > 1.5 mg/dL. ^†^ Chi-square test; ^‡^ Mann–Whitney U test; ^#^ Fisher’s exact test. Continuous data were expressed as median and range. Categorical data were expressed as number and percentage.

**Table 2 jcm-10-05114-t002:** Treatment comparisons among lymphoma-associated HLH patients diagnosed in 2004–2012 and 2013–2021.

	Total		Year of Diagnosis 2004–2012		Year of Diagnosis 2013–2021		*p* Value
**Treatment of B-cell lymphoma (*n* = 29)**							
Without rituximab	8	(27.6%)	5	(38.5%)	3	(18.8%)	0.406
With rituximab	21	(72.4%)	8	(61.5%)	13	(81.2%)	
**Treatment of B-cell lymphoma (*n* = 29)**							
Best supportive care	4	(13.8%)	1	(7.7%)	3	(18.8%)	0.606
Intent-to-cure chemotherapy	25	(86.2%)	12	(92.3%)	13	(81.2%)	
**Treatment of T-cell lymphoma (*n* = 27)**							
Best supportive care	8	(29.6%)	4	(23.5%)	4	(40.0%)	0.415
Intent-to-cure chemotherapy	19	(70.4%)	13	(76.5%)	6	(60.0%)	

HLH: hemophagocytic lymphohistiocytosis. This result is analyzed by the Fisher’s exact test. All the data are shown as number and percentage.

**Table 3 jcm-10-05114-t003:** Comparison of treatment outcome by first line intent-to-cure therapy.

	Total (*n* = 56)		Diagnosis in 2004–2012 (*n* = 30)		Diagnosis in 2013–2021 (*n* = 26)		*p* Value
**Response *, *n* (%)**							0.137 ^†^
CR	21	(55.3%)	11	(50.0%)	10	(62.5%)	
PR	4	(10.5%)	1	(4.5%)	3	(18.8%)	
PD	13	(34.2%)	10	(45.5%)	3	(18.8%)	
**Death, *n* (%)**							0.353 ^†^
No	15	(26.8%)	6	(20.0%)	9	(34.6%)	
Yes	41	(73.2%)	24	(80.0%)	17	(65.4%)	

CR: complete remission; PR: partial response; SD: stable disease; PD: progressive disease. ^†^ Chi-square test. All data are shown as number and percentage. * 18 patients died before treatment response assessment.

**Table 4 jcm-10-05114-t004:** Prognostic factors of lymphoma-associated HLH.

	Univariate Analysis			Multivariate Analysis		
	HR	(95% CI)	*p* Value	HR	(95% CI)	*p* Value
**Age, years**						
≤60	1.00					
>60	1.40	(0.72–2.75)	0.324			
**Gender**						
Male	1.00					
Female	0.90	(0.45–1.79)	0.757			
**Lymphoma subtype**						
B-cell lymphoma	1.00					
T-cell lymphoma	1.17	(0.60–2.29)	0.650			
**ECOG Performance status**						
1–2	1.00			1.00		
3–4	4.19	(2.08–8.42)	<0.001	5.38	(2.49–11.61)	<0.001
**Stage**						
Stage 1, 2	1.00					
Stage 3, 4	1.50	(0.36–6.26)	0.580			
**Ferritin, ng/mL**						
<3000	1.00					
≥3000	1.44	(0.68–3.07)	0.339			
**LDH, IU/L**						
<1000	1.00					
≥1000	1.17	(0.60–2.31)	0.645			
**DIC**						
No	1.00			1.00		
Yes	2.06	(1.01–4.19)	0.046	2.08	(0.99–4.40)	0.054
**Jaundice**						
No	1.00			1.00		
Yes	2.78	(1.34–5.76)	0.006	2.91	(1.37–6.18)	0.006
**IPI**						
0–3	1.00					
4–5	1.74	(0.87–3.48)	0.118			
**Year of diagnosis**						
2013–2021	1.00					
2004–2012	1.41	(0.70–2.81)	0.336			

HLH: hemophagocytic lymphohistiocytosis; ECOG: Eastern Cooperative Oncology Group; LDH: lactate dehydrogenase; DIC: disseminated intravascular coagulation; IPI: international prognostic index; HR: hazard ratio; CI: confidence interval. Jaundice is defined when serum total bilirubin >1.5 mg/dL.

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
