# Peer review of "A Decade of Lymphoma-Associated Hemophagocytic Lymphohistiocytosis: Does the Outcome Improve?"

_jcm, 2021, doi:10.3390/jcm10215114_

Round 1
Reviewer 1 Report
The paper is elegantly written and reveals a clear message, namely that lymphoma-associated HLH has a dismal outcome that cannot be significantly influenced in a second series of lymphoma-associated HLH patients by more modern therapeutic managements during a later time-period compared to an earlier cohort. However, what I am wondering about is how the authors can in general interprete these negative comparative results concerning the outcome of secondary HLH while treatment strategies mainly target a variety of underlying malignant lymphomas.
Author Response
Thanks for this critical question. Although our study cohort comprised a variety of lymphoma subtypes, the subtypes of lymphomas in these two cohorts were not significantly different (Table1). Besides, the regimens between these two cohorts were quite comparable (Supplemental Table 2). Taking these data together, we proposed that early diagnosis and tailored treatments that balance efficacy and adverse events remain the key to obtaining a better outcome in lymphoma-associated HLH.
Reviewer 2 Report
The authors retrospectively investigated the potential treatment evolution and outcome improvement in patients diagnosed and treated with lymphoma-associated HLH between 2004–2012 (n = 30) and 2013–2021 (n = 26). With a comparable response rate of lymphoma treatment, patients treated in 2013–2021 had a numerically greater median OS than those treated in 2004–2012 (23.6 ± 19.8 vs 9.7 ± 4.5 months).
How many patients undergo hematopoietic stem cell transplantation (HSCT) following chemotherapy?
It has been reported that PET‑CT may act as a significant tool to assess patients with lymphoma-associated HLH, as it is highly sensitive in detecting neoplasms of the majority of histologic subtypes of lymphoma. The authors should discuss the issue.
Is there any difference between patients with T/NK‑cell lymphoma-associated HLH compared with patients with B-cell lymphoma-associated HLH in plasma EBV DNA?
The authors should provide a therapeutic algorithm according to their experience.
Author Response
Reviewer 2
1. How many patients undergo hematopoietic stem cell transplantation (HSCT) following chemotherapy?
Response:
Thanks for this important question. Six patients in our study underwent allo-HSCT. We summarized pre- and post-transplant disease status, donor types, conditioning regimens, and the current status of these six patients (Supplemental Table 1; page 4).
2. It has been reported that PET CT may act as a significant tool to assess patients with lymphoma-associated HLH, as it is highly sensitive in detecting neoplasms of the majority of histologic subtypes of lymphoma. The authors should discuss the issue.
Response:
Appreciate! We have followed the suggestion, adding the description of potential role of PET/CT on early diagnosis of lymphoma-associated HLH (page 7).
3. Is there any difference between patients with T/NK cell lymphoma-associated HLH compared with patients with B-cell lymphoma-associated HLH in plasma EBV DNA?
Response:
Great question! The plasma EBV DNA in lymphoma patients were not routinely checked in our daily practice. We have added this limitation to our revised manuscript (page 8).
4. The authors should provide a therapeutic algorithm according to their experience.
Response:
Thanks for the great suggestion. We have followed this suggestion, adding a therapeutic algorithm as a supplemental figure (page 7).